# Proposed Cut-Off Score for the Japanese Version of the Fear of Coronavirus Disease 2019 Scale (FCV-19S): Evidence from a Large-Scale National Survey in Japan

**DOI:** 10.3390/ijerph20010429

**Published:** 2022-12-27

**Authors:** Haruhiko Midorikawa, Hirokazu Tachikawa, Miyuki Aiba, Yuki Shiratori, Daichi Sugawara, Naoaki Kawakami, Ryo Okubo, Takahiro Tabuchi

**Affiliations:** 1Department of Psychiatry, University of Tsukuba Hospital, 2-1-1 Amakubo, Tsukuba 305-8575, Japan; 2Department of Disaster and Community Psychiatry, Division of Clinical Medicine, Faculty of Medicine, University of Tsukuba, 1-1-1 Tennoudai, Tsukuba 305-8575, Japan; 3Faculty of Human Sciences, Toyo Gakuen University, 1-26-3 Hongo, Bunkyo, Tokyo 113-0033, Japan; 4Department of Psychiatry, Division of Clinical Medicine, Faculty of Medicine, University of Tsukuba, 1-1-1 Tennoudai, Tsukuba 305-8575, Japan; 5Faculty of Human Sciences, University of Tsukuba, 1-1-1 Tennoudai, Tsukuba 305-8575, Japan; 6Department of Neuropsychiatry, National Hospital Organization Obihiro Hospital, 18-2-16, Obihiro 080-8518, Japan; 7Cancer Control Center, Osaka International Cancer Institute, 3-1-69 Otemae, Chuo-ku, Osaka 541-8567, Japan

**Keywords:** COVID-19, mental health, fear, depression, psychological scale

## Abstract

The Fear of coronavirus disease 2019 (COVID-19) Scale (FCV-19S) is a seven-item self-administered psychological scale for measuring the fear of this disease. The scale has been widely adapted and validated worldwide. This study aimed to propose a cut-off score for the validated Japanese version of the FCV-19S. We conducted a nationwide online survey and included 26,286 respondents in the analysis. Respondents answered questions on their sociodemographic characteristics, and using the FCV-19S and six-item Kessler Psychological Distress Scale, we measured psychological distress and assessed whether the fear of COVID-19 interfered with their daily lives. A total score of ≥21 points was considered adequate to identify those with psychological distress or difficulties in daily living because of the fear of COVID-19. This cut-off score will contribute to mental health assessment during the COVID-19 pandemic.

## 1. Introduction

The coronavirus disease 2019 (COVID-19) pandemic continues to spread globally, with more than 400 million cases reported in over 200 countries/regions worldwide as of March 2022 [1]. Japan has experienced six waves of the COVID-19 pandemic [2], and the number of infected patients has exceeded 5 million [2]. The pandemic has resulted in a variety of mental health issues. The problem was initially a temporary mental health deterioration due to lockdown [3]; however, ongoing effects, such as an increase in the prevalence of depression and anxiety disorders and the number of suicides [4], have become problematic [5]. Factors associated with mental health during the COVID-19 pandemic range from individual-level factors, such as sex, age, and socioeconomic status [6], to population-level factors, such as reduced social interaction (COVID-19 Mental Disorders Collaborators 2021).

Among the factors related to mental health, fear and anxiety are the most distinctive and essential during the COVID-19 pandemic. Fear of COVID-19 has been associated with severe psychological distress [7], while some studies have associated it with health-promoting behaviors [8]. Hence, measuring the fear and anxiety of individuals during the COVID-19 pandemic is essential. For this purpose, various scales have been developed, such as the Fear of COVID-19 Scale (FCV-19S) [9], the Coronavirus Anxiety Scale [10], and the 36-item COVID Stress Scale [11].

The FCV-19S is a seven-item self-administered psychological scale [9] used to measure the fear of COVID-19. It is easily measurable and has been adapted in various countries worldwide, such as Japan [12,13,14], China [15], and Bangladesh [16]. In Europe, it has been used in France [17], Italy [18], Spain [19], Greece [20], Turkey [21], Norway [22], and Romania [23]. There are also reports of its use in the United States [24] and Brazil [25]. Some studies have reported a one-factor structure [26], while others have used a two-factor structure (emotional fear reactions and symptomatic expression of fear) [12,14]. However, reports on its cut-off value have been limited to the Greek version [27], and the cut-off value has not been examined in other languages, countries, or periods. A previous study [27] proposed a cut-off value using scales, such as the Generalized Anxiety Disorder -7 (GAD-7), Short Health Anxiety Inventory, and posttraumatic stress disorder (PTSD)-8 scale. However, these scale scores were elevated by other factors and did not consider whether fear of COVID-19 interfered with daily life. Moreover, the cut-off values of the FCV-19S scores were not examined by factors, assuming FCV-19S as two factors.

As the FCV-19S is a psychological scale that measures the fear of COVID-19, in determining the cut-off value, it should screen out those individuals whose daily lives are disrupted by the fear of COVID-19. Moreover, the FCV-19S would be better at screening those whose fear of COVID-19 is interfering with their daily lives than those with psychological distress that is also affected by various factors other than fear of COVID-19. In contrast, another point that should be elucidated is whether the optimal cut-off values for FCV-19S differ significantly in screening for these problems. Further, given that the FCV-19S has been reported to have a two-factor structure, it is necessary to consider the possibility that screening for these problems may be better using factor-specific scores rather than the total scores. In addition, setting a high sensitivity cut-off value for screening will reduce specificity, but the balance between sensitivity and specificity should be considered given the wide variety of applications for the FCV-19S.

Therefore, we conducted this study using data from a large national survey in Japan to examine cut-off values for the FCV-19S regarding daily life disturbances due to the fear of COVID-19 and psychological distress, to characterize the cut-off values for the FCV-19S total score and scores by factor, and to examine the factors associated with scores higher than the cut-off values.

## 2. Materials and Methods

### 2.1. Study Design and Participants

This was a cross-sectional study using the Japan COVID-19 and Society Internet Survey (JACSIS) data, a nationally representative survey. The JACSIS was launched in 2020 to investigate how social issues, such as health, medical care, work styles, and the economy, have changed during the COVID-19 pandemic in Japan. The survey panel was from approximately 2.2 million panelists at a Japanese Internet research company (Rakuten Insight, Inc., Tokyo, Japan) and comprised individuals from diverse socioeconomic backgrounds, such as educational level, household income, number of household members, and marital status [28]. We analyzed data from the JACSIS in 2021, the second survey of this project. In this second survey carried out from 27 September 2021 to 29 October 2021, we distributed questionnaires to 33,081 candidates who answered the first survey. In total, 69% of the participants (n = 22,838) responded to the survey. The survey continued for the new panelists until the number of respondents reached the targeted sample size (n = 31,000). The participants who provided informed consent on the web provided information on their sociodemographic characteristics and mental health and were free to stop responding at any time. Of these, we included 26,286 in the final analysis (84.7% of all survey respondents) after excluding unnatural or inconsistent responses using filler questions and responses from people aged < 20 years.

### 2.2. Measures

The sociodemographic characteristics of participants included age, education, marital status, cohabitation, occupation, and income. We measured the mental status of the individuals using the Japanese version of the Kessler Psychological Distress Scale (K6) and the Japanese version of the FCV-19S. The participants were also asked how the fear of COVID-19 has affected their work, care of their home, or interaction with other people (difficulties in daily living due to the fear of COVID-19). The question on the interference with daily life corresponded to the phrase “the disorder is causing clinically meaningful distress or impairment in social, occupational, or other important areas of functioning” used in the DSM-5 to diagnose mental disorders [29]. This question is also used in psychological scales, such as the Patient Health Questionnaire-9 (PHQ-9) [30], GAD-7 [31], and Moral Injury Symptom Scale–Healthcare Professionals version [32].

The K6 is a six-item self-administered psychological scale with items measured on a 5-point Likert scale ranging from 0 points, indicating “never” to 4 points indicating “all the time.” This scale was developed to screen mood and anxiety disorders [33]. K6 is typically considered a one-factor model and total K6 scores can range from 0 to 24 points [33,34]. There is evidence of validity and reliability that supports the use of K6 in the Japanese population [34]. Based on the recommendations of previous studies [35,36], K6 ≥ 5 points was adopted as the cut-off value to determine whether the individual is in moderate or higher psychological distress.

In the FCV-19S, respondents were asked to indicate their level of agreement with each question on a scale of 1 (strongly disagree) to 5 (strongly agree) points. The total FCV-19S scores ranged from 7 to 35 points. Assuming a two-factor model (Factor 1: emotional fear reactions, Factor 2: symptomatic expression of fear), Factor 1 is the sum of the following four questions: Question 1 (“I am most afraid of COVID-19”), Question 2 (“It makes me uncomfortable to think about COVID-19”), Question 4 (“I am afraid of losing my life because of COVID-19”), and Question 5 (“When watching news and stories about COVID-19 on social media, I become nervous or anxious”); the scores ranged from 4 to 20 points. Factor 2 is the sum of the following three questions: Question 3 (“My hands become clammy when I think about COVID-19”), Question 6 (“I cannot sleep because I am worried about getting COVID-19”), and Question 7 (“My heart races or palpitates when I think about getting COVID-19”); the scores ranged from 3 to 15 points. There is evidence of validity and reliability that supports the use of FCV-19S in the Japanese population [14]. The confirmatory reliability analysis in that previous study showed that the Cronbach’s alpha for the Japanese version of FCV-19S was 0.83 in the one-factor model (χ^2^ = 386.25, *p* < 0.001, comparative fit index [CFI]: 0.979, Tucker–Lewis index [TLI]: 0.944, root mean square error of approximation [RMSEA]: 0.084, and Akaike’s information criterion [AIC]: 426.25), as well as 0.77 for Factor 1 and 0.83 for Factor 2 in the two-factor model (χ^2^ = 164.15, *p* < 0.001, CFI: 0.991, TLI: 0.985, RMSEA: 0.043, AIC: 196.15) [14].

### 2.3. Statistical Analysis

Descriptive statistical summary of the individuals’ sociodemographic characteristics and K6 screening results, whether they had daily living difficulties due to the fear of COVID-19, and their total FCV-19S scores in each attribute are presented according to sex using means and standard deviations (SDs). We conducted confirmatory factor analyses (CFA) of one-factor and two-factor models with maximum likelihood estimation to verify the internal structure of FCV-19S considering as good goodness of fit indexes of CFI ≥ 0.95, TLI ≥ 0.95, RMSEA ≤ 0.06, and AIC. Reliability was evaluated using Cronbach’s alpha. A receiver operating characteristic (ROC) curve analysis was performed to examine the cut-off values useful for distinguishing whether the fear of COVID-19 interfered with daily life and whether the participants experienced moderate or severe psychological distress (K6 ≥ 5 points) for each of the three scores (FCV-19S total score, FCV-19S Factor 1 total score, and FCV-19S Factor 2 total score). Youden’s Index [37] was calculated to determine the optimal cut-off value. Finally, we divided the respondents based on the determined cut-off value of FCV-19S. We examined the association between each demographic characteristic and the fear of COVID-19 using the chi-squared test and effect size (Cramer’s V) in total sample. Statistical significance was set at *p* < 0.05. An effect size of 0.1 or greater was considered to indicate a small effect. We performed statistical analyses using IBM SPSS Statistics for Windows, Version 27.0 (IBM Corp., Armonk, NY, USA). CFA was performed with the statistical package IBM Amos for Windows, Version 26.

## 3. Results

Of the 26,286 individuals, 12,732 (48.4%) were men, 11,794 (44.9%) were between the ages of 40 and 64 years, 14,716 (56.0%) had graduated from a university or junior college, 16,265 (61.9%) were married, and 5469 (20.8%) lived alone. Those in moderate or severe psychological distress (K6 ≥ 5 points) accounted for approximately 1/3 of the participants, and one in six-to-seven individuals had problems in daily life due to the fear of COVID-19. The sociodemographic data are presented in Table 1. The mean total FCV-19S score for the entire cohort was 18.3 (SD = 5.1), with 17.9 (SD = 5.3) for men and 18.7 (SD = 4.9) for women. Women with moderate or severe psychological distress or those with problems in daily life due to the fear of COVID-19 had total FCV-19S scores >20 points.

The results of the CFA model fit are presented in Table 2. All indices indicate a good fit for both models. Reliability analysis assessing both models indicated a Cronbach’s alpha of 0.84 for the one-factor model, as well as 0.81 for Factor 1 and 0.90 for Factor 2.

The results of the ROC curve analysis are presented in Table 3 and Table 4 and Figure 1 and Figure 2. The FCV-19S total score significantly predicted whether the patient had problems in daily life due to the fear of COVID-19 (area under the curve [AUC]: 0.751, 95% confidence interval [CI]: 0.742–0.759, *p* < 0.001, sensitivity: 0.630, specificity: 0.728). The optimal cut-off value was 21 points (Youden’s index: 0.359). The FCV-19S Factor 1 score significantly predicted participants with problems in daily life due to the fear of COVID-19 (AUC: 0.747, 95% CI: 0.739–0.756, *p* < 0.001, sensitivity: 0.624, specificity: 0.744). The optimal cut-off value was 15 points (Youden’s index: 0.369). The FCV-19S Factor 2 score significantly predicted the abovementioned participants (AUC: 0.669, 95% CI: 0.659–0.679, *p* < 0.001, sensitivity: 0.503, specificity: 0.764). The optimal cut-off value was 7 points (Youden’s index: 0.267). The FCV-19S total score significantly predicted whether the patient had moderate or severe psychological distress (AUC: 0.645, 95% CI: 0.638–0.652, *p* < 0.001, sensitivity: 0.388, specificity: 0.817). The optimal cut-off value was 21 points (Youden’s index: 0.226). The FCV-19S Factor 1 score significantly predicted the abovementioned participants (AUC: 0.594, 95% CI: 0.587–0.602, *p* < 0.001, sensitivity: 0.404, specificity: 0.737). The optimal cut-off value was 15 points (Youden’s index: 0.140). The FCV-19S Factor 2 score significantly predicted the abovementioned participants (AUC: 0.645, 95% CI: 0.637–0.652, *p* < 0.001, sensitivity: 0.438, specificity: 0.808). The optimal cut-off value was 7 points (Youden’s index: 0.247).

The relationship between the cut-off values and the sensitivity and specificity of each FCV-19S score are shown in Table 5 and Table 6. The cut-off values with the largest Youden’s Index are highlighted in italics in the tables.

Table 7 shows the relationship between each variable and FCV-19S when the cut-off value was set at 20.5 points. Among the total individuals, 8584 (32.7%) had a score of 21 or higher. Significant differences were found for all variables, but the effect sizes were small (>0.1).

## 4. Discussion

This study had three objectives. The first objective was to propose a cut-off value for FCV-19S using data from a large national survey in Japan. As cut-offs are often used in psychiatry and psychology to detect mood and anxiety disorders, such as the PHQ-9 and GAD-7 [30], using cut-offs to distinguish individuals is important for comparing and interpreting the study results. However, cut-off values of the FCV-19S have not been thoroughly examined, and many previous studies using the FCV-19S have divided individuals by medians or percentiles [7,38]. In this study, the cut-off value of the FCV-19S was examined using two approaches: the K6, a widely used measure of psychological distress; and a question regarding whether the fear of COVID-19 itself interferes with daily life. A study using the former was conducted by Nikopoulou et al. [27]. They proposed a cut-off value for the FCV-19S based on its relevance to existing psychological scales. They showed that a cut-off value of 16.5 points significantly predicted anxiety, health anxiety, and post-traumatic symptomatology. While this is an important finding in examining the cut-off value of the FCV-19S, the anxiety, health anxiety, and post-traumatic stress symptoms measured by the psychological scale in that study were not necessarily caused by the fear of COVID-19. In addition, as they discussed in their study, the cut-off value of 16.5 points was below the average score in many studies conducted during the pandemic’s peak and below the average score in this study. However, as the scale was designed to measure the fear of COVID-19, it is better to have a cut-off value that can screen a small group of people whose fear of COVID-19 interferes with their daily lives, compared to a cut-off value that would apply to most people.

Based on the results of this study, we propose a cut-off value of 21 points for the FCV-19S total score. Based on the AUC results, this cut-off value is moderately accurate in distinguishing between daily life disturbances due to the fear of COVID-19. The optimal cut-off value with the highest Youden’s index was the same concerning daily life disturbance due to the fear of COVID-19 and concerning psychological distress. However, as expected, since the scale consists of questions related to fear of COVID-19, it was not as good at distinguishing between the presence of psychological distress as it was at distinguishing between the presence of daily life disturbances caused by fear of COVID-19. It is reasonable to state that the FCV-19S is more useful in distinguishing those with troubles in daily life due to the fear of COVID-19 than in identifying those with psychological distress, as the causes of the psychological distress measured in this study were not limited to the fear of COVID-19. In the future, if the risk of COVID-19 is further reduced through the development of more effective vaccines and treatments, fear of COVID-19 will account for a smaller proportion of people’s psychological distress, thus, making it more difficult to use the FCV-19S for screening for psychological distress. In contrast, if the number of people whose fear of COVID-19 interferes with their daily lives is reduced from the point at which this study was conducted, the significance of identifying high-risk individuals for fear of COVID-19 through screening will increase because more focused measures can be taken for these individuals.

The second objective of this study was to characterize the cut-off values for the FCV-19S total score and each of the scores by factor. Comparison of the FCV-19S total score and the cut-offs for the FCV-19S factor scores yielded different results depending on the event being distinguished. Regarding the daily life obstacles caused by the fear of COVID-19, Factor 2 “Symptomatic expression of fear” was less accurate than the total score and Factor 1 “Emotional fear reactions.” Conversely, Factor 1 was less accurate than the total score, while Factor 2 distinguished psychological distress. Factor 1 of the FCV-19S includes items on emotional reactions to fear, while Factor 2 includes items on physical symptoms caused by fear. The distribution of scores by factor also differed, as the mode for Factor 1 was closer to the mean of the data than that of Factor 2, while the mode for Factor 2 was the minimum score [14,18]. These results suggest that it may be helpful to refer not only to the total FCV-19S score but also to each score by factor when ascertaining the details of the individual‘s fear of COVID-19. For example, in diseases, such as PTSD, which can cause physical symptoms as a physiological response [39], it may be necessary to give more weightage to Factor 2 scores than to Factor 1 scores. However, as the total score was a good indicator of the obstacles in daily life due to COVID-19 and psychological distress for which cut-off values were considered in this study, it was considered unnecessary to use factor-specific scores.

The third objective of this study was to examine the factors associated with scores higher than the cut-off values. Sex, age, education, marital status, cohabitation, occupation, and income were all associated with the percentage of those whose scores exceeded the FCV-19S cut-off value; however, the effect sizes were small. The associations found in this study were generally consistent with those of previous studies [14,24]. In these previous studies, being at risk of infection, having been infected, and having been disturbed by the self-confinement associated with COVID-19 were also associated with higher FCV-19S scores than the demographic variables. The results of this study and those of previous studies suggest that more effective measures to reduce fear of COVID-19 could be taken by utilizing FCV-19S screening to narrow the target population rather than targeting specific attributes, such as age and sex.

To address mental health issues during the COVID-19 pandemic, it is important to consider the fear of COVID-19. For instance, the screening of those who plan to engage in work directly related to patients with COVID-19, which mainly applies to healthcare workers, and determining if they have an intense fear of COVID-19, measures such as providing support to reduce their fear or preventing them from engaging in such work could be undertaken. It is also possible to longitudinally follow a population with an intense fear of COVID-19 to identify the characteristics of individuals with a persistent fear of COVID-19 and to examine effective countermeasures. In this study, a cut-off value that is the best balance between sensitivity and specificity was set using Youden’s index so that the FCV-19S can be widely used for various initiatives. Although it is permissible to change the cut-off value depending on the purpose of its use, the current cut-off value is a score that is reached even if the respondent answers “3. Neither agree nor disagree” to all questions. Here, as approximately 30% of the population was included in this category, lowering the cut-off value to increase further the sensitivity would reduce the significance of narrowing down the target population. In contrast, further increasing the cut-off value would make it difficult to grasp the whole picture of a population with an intense fear of COVID-19. In addition, although this cut-off is not intended to diagnose disease, attention should also be paid to false negatives. Specifically, it is essential to provide continuous mental health care to those who put themselves in a position where fear of COVID-19 is an issue, even if they are below the cut-off value, in anticipation of the possibility that some may not receive an appropriate response despite their fear of COVID-19 because they do not match the screening criteria.

The fact that the cut-off values differed from those of previous studies may be attributed to various factors, including the timing of the study, the population targeted, and differences in cultural backgrounds. Therefore, although the FCV-19S has been adapted into multiple languages, it is advisable to examine its validity in each country or region when using the cut-off values.

This study had several limitations. First, although this was a large-scale survey conducted on a national scale, it was a web-based online questionnaire survey; hence, the effect of selection bias could not be ruled out. Second, as the survey was conducted in Japanese and is considered to have been answered primarily by Japanese individuals, care must be taken when adapting the results of this study to other countries. Finally, as only the K6 was used as the psychological scale, we could not examine the relationship between the FCV-19S and scales measuring anxiety and PTSD.

## 5. Conclusions

In conclusion, a total score of 21 points was a reasonable cut-off value for distinguishing populations, in which the fear of COVID-19 was a problem. In addition, various sociodemographic characteristics were associated with the FCV-19S cut-off value. Using the FCV-19S cut-off value to identify people with the fear of COVID-19 is important for mental health measures during the COVID-19 pandemic. As there is still a lack of research on cut-off values for the FCV-19S, more countries and regions should consider cut-off values in the national language versions of the FCV-19S. While establishing cut-off values for diagnostic purposes, it would also be desirable to conduct studies that measure the FCV-19 scores in individuals with a diagnosis of specific phobia adapted for COVID-19 in the DSM-5.

## Figures and Tables

**Figure 1 ijerph-20-00429-f001:**
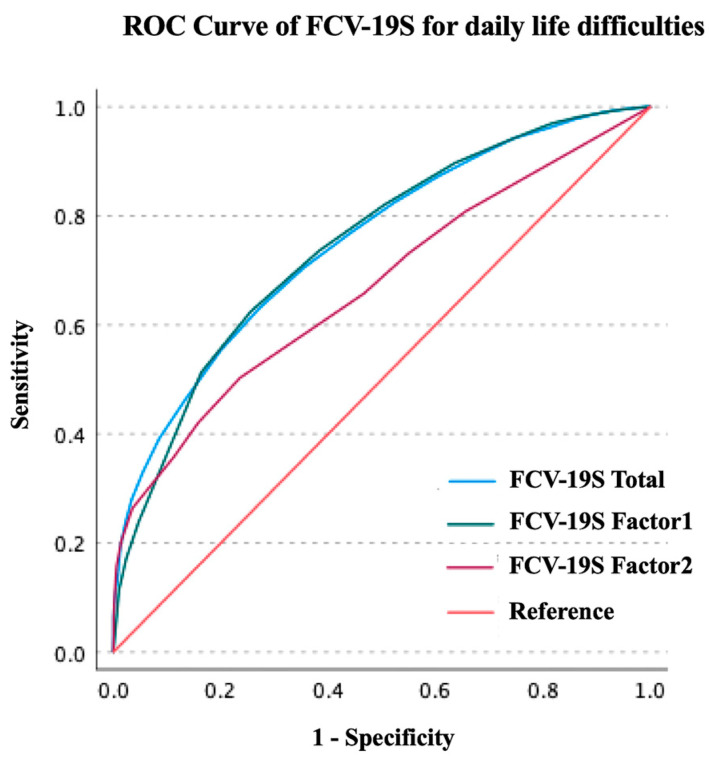
ROC curve of FCV-19S for daily life difficulties due to the fear of COVID-19. FCV-19S, Japanese Version of the Fear of COVID-19 Scale; ROC, receiver operating characteristic; COVID-19, coronavirus disease 2019.

**Figure 2 ijerph-20-00429-f002:**
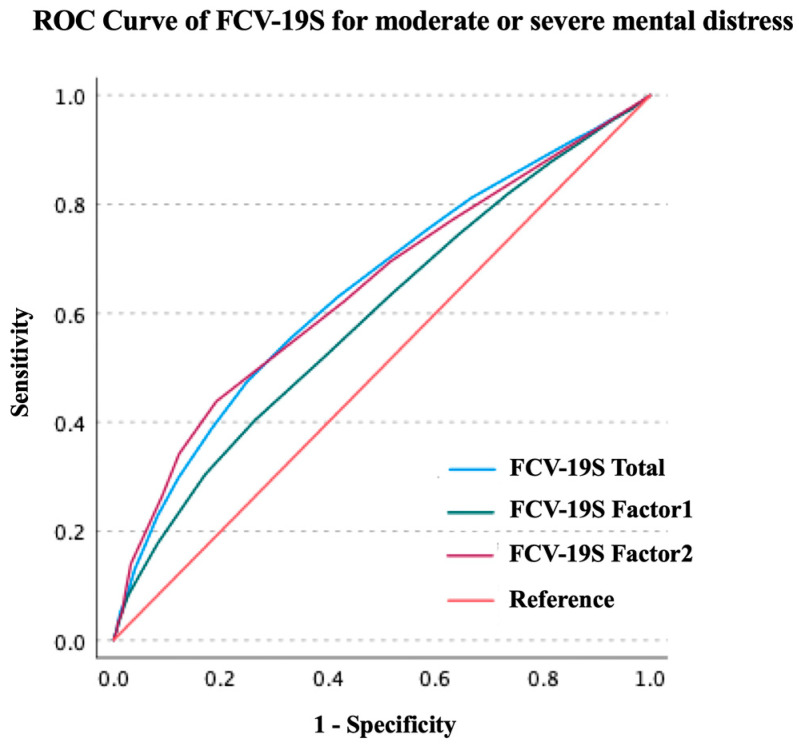
ROC curve of FCV-19S for moderate or severe psychological distress. FCV-19S, Japanese Version of the Fear of COVID-19 Scale; ROC, receiver operating characteristic.

**Table 1 ijerph-20-00429-t001:** Characteristics of study participants and FCV-19S scores for each attribute.

Characteristics	Characteristics Name	n (%)	Women		Men	
			Mean	SD	Mean	SD
Age group (years)	20–39	7321 (27.9)	18.50	5.10	17.69	5.64
	40–64	11,794 (44.9)	18.70	4.96	17.76	5.40
	65–79	7171 (27.3)	19.05	4.51	18.40	4.69
Education	≤High School Unknown	11,570 (44)	18.93	4.89	18.13	5.46
	>High School	14,716 (56)	18.56	4.87	17.78	5.20
Marital status	Single	7254 (27.6)	18.16	5.14	17.66	5.69
	Divorced Bereaved	2767 (10.5)	18.70	4.85	17.35	5.33
	Married	16,265 (61.9)	18.99	4.77	18.09	5.09
Cohabitant	No	5469 (20.8)	18.02	5.03	17.36	5.60
	Yes	20,817 (79.2)	18.91	4.83	18.06	5.21
Occupation	Self-employment	1799 (6.8)	17.86	5.16	17.61	5.41
	Part-time job	5177 (19.7)	18.76	4.88	17.92	5.31
	Employed	9996 (38)	18.37	5.13	17.80	5.39
	Unemployed	9314 (35.4)	19.02	4.70	18.26	5.05
Income	<2,000,000 JPY	2225 (8.5)	18.86	5.27	18.48	6.31
	≥2,000,000– <6,000,000 JPY	10,555 (40.2)	18.60	4.77	18.03	5.17
	≥6,000,000– <12,000,000 JPY	6771 (25.8)	18.39	4.76	17.51	5.11
	≥12,000,000 JPY	1372 (5.2)	17.95	5.37	16.84	5.11
	Other/Unknown	5363 (20.4)	19.36	4.87	18.53	5.46
K6	≥5	9094 (34.6)	20.34	5.21	19.60	5.52
	<5	17,192 (65.4)	17.85	4.45	17.06	4.98
Problems due to the fear of COVID-19	Yes	4022 (15.3)	22.58	5.39	22.68	5.86
	No	22,264 (84.7)	17.99	4.40	17.13	4.77

**Table 2 ijerph-20-00429-t002:** Model fit indices of CFA.

	χ^2^	(df)	*p*	CFI	TLI	RMSEA	AIC
One-factor	382.98	8	<0.001	0.996	0.990	0.042	422.98
Two-factor	713.57	10	<0.001	0.993	0.984	0.052	749.57

CFI, comparative fit index; TLI, Tucker–Lewis index; RMSEA, root mean square error of approximation; AIC, Akaike’s information criterion.

**Table 3 ijerph-20-00429-t003:** Predictive validity of FCV-19S for problems due to the fear of COVID-19.

FCV-19S Type	Cut-Off	Sensitivity	Specificity	AUC	95% CI	Significance	Youden’s Index
FCV-19S Total	21	0.630	0.728	0.751	0.742–0.759	<0.001	0.359
FCV-19S Factor 1	15	0.624	0.744	0.747	0.739–0.756	<0.001	0.369
FCV-19S Factor 2	7	0.503	0.764	0.669	0.659–0.679	<0.001	0.267

FCV-19S, Japanese Version of the Fear of COVID-19 Scale; COVID-19, coronavirus disease 2019; CI, confidence interval; AUC, area under the curve.

**Table 4 ijerph-20-00429-t004:** Predictive validity of FCV-19S in moderate or severe psychological distress.

FCV-19S Type	Cut-Off	Sensitivity	Specificity	AUC	95% CI	Significance	Youden’s Index
FCV-19S Total	21	0.388	0.817	0.645	0.638–0.652	<0.001	0.226
FCV-19S Factor 1	15	0.404	0.737	0.594	0.587–0.602	<0.001	0.140
FCV-19S Factor 2	7	0.438	0.808	0.645	0.637–0.652	<0.001	0.247

FCV-19S, Japanese Version of the Fear of COVID-19 Scale; CI, confidence interval; AUC, area under the curve.

**Table 5 ijerph-20-00429-t005:** Sensitivity and specificity for the selection of cut-off points for problems due to the fear of COVID-19.

FCV-19S Total			FCV-19S Factor 1			FCV-19S Factor 2		
Cut-off	Sensitivity	Specificity	Cut-off	Sensitivity	Specificity	Cut-off	Sensitivity	Specificity
13	0.977	0.140	7	0.995	0.051	4	0.808	0.344
14	0.962	0.186	8	0.992	0.076	5	0.731	0.450
15	0.942	0.254	9	0.981	0.135	6	0.658	0.533
16	0.911	0.319	10	0.969	0.185	*7*	*0.503*	*0.764*
17	0.872	0.396	11	0.938	0.266	8	0.421	0.842
18	0.825	0.476	12	0.897	0.363	9	0.358	0.887
19	0.768	0.559	13	0.821	0.494	10	0.264	0.965
20	0.705	0.646	14	0.736	0.617	11	0.203	0.985
*21*	*0.630*	*0.728*	*15*	*0.624*	*0.744*	12	0.156	0.994
22	0.553	0.800	16	0.513	0.836	13	0.090	0.998
23	0.461	0.868	17	0.316	0.920	14	0.070	0.999
24	0.392	0.915	18	0.239	0.953	15	0.058	1.000

FCV-19S, Japanese Version of the Fear of COVID-19 Scale; COVID-19, coronavirus disease 2019.

**Table 6 ijerph-20-00429-t006:** Sensitivity and specificity for the selection of cut-off points for moderate or severe psychological distress.

FCV-19S total			FCV-19S Factor 1			FCV-19S Factor 2		
Cut-off	Sensitivity	Specificity	Cut-off	Sensitivity	Specificity	Cut-off	Sensitivity	Specificity
13	0.918	0.143	7	0.966	0.050	4	0.771	0.370
14	0.891	0.192	8	0.952	0.075	5	0.695	0.484
15	0.851	0.263	9	0.912	0.132	6	0.622	0.571
16	0.811	0.334	10	0.879	0.183	*7*	*0.438*	*0.808*
17	0.755	0.413	11	0.820	0.264	8	0.342	0.878
18	0.694	0.496	12	0.743	0.359	9	0.264	0.910
19	0.630	0.583	13	0.631	0.487	10	0.141	0.967
20	0.555	0.669	14	0.520	0.607	11	0.085	0.978
*21*	*0.474*	*0.751*	*15*	*0.404*	*0.737*	12	0.052	0.984
22	0.388	0.817	16	0.304	0.829	13	0.029	0.992
23	0.298	0.879	17	0.178	0.917	14	0.020	0.993
24	0.228	0.918	18	0.124	0.949	15	0.016	0.994

FCV-19S, Japanese Version of the Fear of COVID-19 Scale.

**Table 7 ijerph-20-00429-t007:** Characteristics of the two groups based on FCV-19S.

Characteristics	Characteristics Name	FCV-19S			
		≥21	<21	*p*	Cramer’s V
Sex	Women	4683 (34.6)	8871 (65.4)	<0.001	0.04
	Men	3901 (30.6)	8831 (69.4)		
Age group	20–39	2342 (32)	4979 (68)	<0.001	0.02
	40–64	3794 (32.2)	8000 (67.8)		
	65–79	2448 (34.1)	4,723 (65.9)		
Education	≤High School/Unknown	4053 (35)	7517 (65)	<0.001	0.05
	>High School	4531 (30.8)	10,185 (69.2)		
Marital status	Single	2245 (30.9)	5009 (69.1)	<0.001	0.02
	Divorced/Bereaved	890 (32.2)	1877 (67.8)		
	Married	5449 (33.5)	10,816 (66.5)		
Cohabitant	No	1599 (29.2)	3870 (70.8)	<0.001	0.04
	Yes	6985 (33.6)	13,832 (66.4)		
Occupation	Self-employment	528 (29.3)	1271 (70.7)	<0.001	0.04
	Part time job	1746 (33.7)	3431 (66.3)		
	Employed	3080 (30.8)	6916 (69.2)		
	Unemployed	3230 (34.7)	6084 (65.3)		
Income	<2,000,000 JPY	808 (36.3)	1417 (63.7)	<0.001	0.08
	≥2,000,000– <6,000,000 JPY	3403 (32.2)	7152 (67.8)		
	≥6,000,000– <12,000,000 JPY	1976 (29.2)	4795 (70.8)		
	≥12,000,000 JPY	344 (25.1)	1028 (74.9)		
	Other/Unknown	2053 (38.3)	3310 (61.7)		

FCV-19S, Japanese Version of the Fear of COVID-19 Scale.

## Data Availability

The data presented in this study are available on request from the corresponding author. The data are not publicly available due to privacy or ethical restrictions.

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
