# Peer review of "Proposed Cut-Off Score for the Japanese Version of the Fear of Coronavirus Disease 2019 Scale (FCV-19S): Evidence from a Large-Scale National Survey in Japan"

_ijerph, 2022, doi:10.3390/ijerph20010429_

Round 1
Reviewer 1 Report
This study provides evidence of external criterion-related validity for the FCV-19S test in a large sample of Japanese individuals. One of the strengths it offers in this regard is the sample size (although not representative). With the cut-off points they want to obtain, they fill a gap in the scientific literature that is currently unresolved.
However, I believe that although the study has the potential to be published, at least two major changes need to be made. First, it is necessary to report evidence related to the internal structure (i.e., confirmatory factor analysis) for the factors used as well as to report the internal consistency reliability (Cronbach's alpha or omega) of the scores used. Without this information it is not possible to assess whether the conclusions drawn from the ROC analysis are valid or not.
On the other hand, I think it is also important to introduce both in the introduction and in the discussion the implications of choosing high or low sensitivity or specificity values. What information do we lose by changing the cut-off points?
Below are some suggestions for improvement (minor and major) that I have detected during the review of the manuscript.
- I recommend changing the word “translated” for “adapted” across the manuscript. It is not only about translating but about adapting it to the cultural context.
- Page 3 line 104. Avoid saying that a test is reliable and valid. A test is not valid or reliable. A suggestion could be “ There is evidence of validity and reliability that supports the use of K6 in Japanese population”. Same in line 120 with FCV-19S. See for example Standards for Educational Psychological Testing https://www.testingstandards.net/uploads/7/6/6/4/76643089/standards_2014edition.pdf
- Furthermore, I would suggest that the authors give more details about the other scales as they have done with the FCV: are they Likert scales? how many factors do they have? what kind of evidence of psychometric quality has been given?
- This article gives evidence of validity related to a criterion and then calculates the ROC area. Although the dimensionality of the scale has been studied in previous studies, it would be desirable that this study also perform a confirmatory factor analysis and report on the internal consistency reliability of the scores. The psychometric properties are linked to a specific context and sample, and given the sample size obtained, it would be convenient to report the psychometric quality of the tools used, as recommended, for example, by the latest version of the APA manual. Having internal consistency reliability values could help to discuss the results obtained on sensitivity and specificity.
- Avoid reporting SD using ± (this is usually for standard error). 18.3 ± 5.1 ïƒ 18.3 (SD = 5.1)
- In table 1, use “Mean” instead of “mean”, or just use “M”. Moreover, describe in title or table Note that mean and sd is for FCV-19S
- Figure 1. Provide better image quality for Figure 1. It would be helpful if the cut-off suggested is labelled or highlighted in the figure.
- I think it is necessary for the authors to hypothesize whether they want the tool to have a high sensitivity or a high specificity since, as can be seen in the results, it is complicated to be able to have both. Thus, it would be easier to be able to evaluate whether the results are favorable or not.
- As I understand from the methods the FCV-19S score ranges from 7 to 35, and is calculated from the sum. Therefore, the scores cannot have decimal places. How can it be that the proposed cut point is 20.5 and not 20 or 21? I think it needs to be explained in the methods or at least in the discussion.
- I think it is necessary to discuss the implications of the proposed cut-off points. What are the consequences of not detecting "healthy" people or "sick" people given the values obtained?
Reviewer 2 Report
In this study, the authors propose a cutoff for the FCV-19S bifactor structure and provide empirical support for the fact that this measure is useful in distinguishing those individuals who have problems in daily life due to the fear of COVID-19.
Please find below some suggestions, as well as concerns that need to be addressed:
line 54 _ Please complete with the validation studies of the adapted versions in Norwegian and Romanian
line 87 _ It is not clear what inconsistent answers means
Did you use filler questions?
line 106 _ I suggest to replace "subject" with "individual"
line 120 _ Please complete with information regarding the reliability and construct validity of the previous study, as well as the reliability in the current study.
line 136 _ Must explain if you calculated in total sample or in clinical subsample
Although the aggregate metric of AUC can range from 0.5 to 1, it is well accepted that the larger it is the better. You say that you have moderate values, but in some situations they are a little over 0.5 or 0.6, more precisely between 0.59 and 0.64. The low AUC values ​​obtained should be explained in detail.
The size of the obtained effect reflects no practical significance, that is, the effect is not large enough to be meaningful in the real world. You obtained a statistically significant p value because you have a huge sample and not because there is actually an effect, its value being less than 1.
To correctly interpret the obtained results, and in the discussions make the connection with the previous studies that analyzed these sociodemographic variables in relation to FCV-19
Round 2
Reviewer 1 Report
Thank you very much for allowing me to review the manuscript again. I believe that the authors have satisfactorily incorporated my suggestions. In order to make the psychometric wording more concise, here are some minor details that I think could be improved:
Line 145 ïƒ structural validity of FCV-19S items and compared goodness of fit ïƒ “to verify the internal structre of FCV-19S considering as good goodness of fit indexes of “
The results of the CFA model fit are presented in Table 2. All indices indicated better
fit for both models ïƒ “All indices indicate a good fit for both models”
In Table 1 I think that probably authors want to say “Model 1 – One factor” , Model 2 – 2 factor” instead of factor 1 or factor 2.
In discussion I suggest to change “factor 1” and “factor 2” for a label that elicites the meaning of the scale.
Reviewer 2 Report
Accept as is.
